# Is there a causal relationship between resistin levels and bone mineral density, fracture occurrence? A mendelian randomization study

**Taichuan Xu, Chao Li, Yitao Liao, Yenan Xu, Zhihong Fan, Xian Zhang**[ID]*

Department of Spine, Wuxi Affiliated Hospital of Nanjing University of Chinese Medicine, Wuxi, Jiangsu, China

* wxzy008@njucm.edu.cn

## Abstract

### Background

In a great many of observational studies, whether there is a relevance of resistin levels on bone mineral density (BMD) and fracture occurrence has been inconsistently reported, and the causality is unclear.

### Methods

We aim to assess the resistin levels on BMD and fracture occurrence within a Mendelian randomization (MR) analysis. Exposure and outcome data were derived from the Integrative Epidemiology Unit (IEU) Open genome wide association studies (GWAS) database. Screening of instrumental variables (IVs) was performed subject to conditions of relevance, exclusivity, and independence. Inverse variance weighting (IVW) was our primary method for MR analysis based on harmonized data. Weighted median and MR-Egger were chosen to evaluate the robustness of the results of IVW. Simultaneously, heterogeneity and horizontal pleiotropy were also assessed and the direction of potential causality was detected by MR Steiger. Multivariable MR (MVMR) analysis was used to identify whether confounding factors affected the reliability of the results.

### Results

After Bonferroni correction, the results showed a suggestively positive causality between resistin levels and total body BMD (TB-BMD) in European populations over the age of 60 [β (95%CI): 0.093(0.021, 0.165), $P = 0.011$]. The weighted median [β(95%CI): 0.111(0.067, 0.213), $P = 0.035$] and MR-Egger [β(95%CI): 0.162(0.025, 0.2983), $P = 0.040$] results demonstrate the robustness of the IVW results. No presence of pleiotropy or heterogeneity was detected between them. MR Steiger supports the causal inference result and MVMR suggests its direct effect.

**Data Availability Statement:** The data presented in this study are openly available in the IEU Open GWAS database (https://gwas.mrcieu.ac.uk/), SCALLOP Consortium's online resources (www.

scallop-consortium.com), GEnetic Factors for OSteoporosis (GEFOS) Consortium website (http://www.gefos.org/), UK Biobank (http://www.nealelab.is/uk-biobank), and the Medical Research Council Integrative Epidemiology Unit (MRC-IEU) website (http://www.bristol.ac.uk/integrative-epidemiology/). The authors confirm others will be able to access these data in the same manner as the authors and that they did not have any special access privileges that others would not have. For legal and ethical reasons, the code used to generate the data in the article cannot be shared publicly. Further inquiries can be directed to the Wuxi Hospital of Traditional Chinese Medicine and the Ethics Committee of Wuxi Hospital of Traditional Chinese Medicine (e-mail: wxtcmirb@163.com) or the corresponding author.

**Funding:** This work was supported by the Project of Jiangsu Provincial Administration of Traditional Chinese Medicine (MS2021044)(http://wjw.jiangsu.gov.cn/col/col57216/index.html)(XZ)and the Scientific Research Project of Wuxi Municipal Health Commission (Z202020)(http://wjw.wuxi.gov.cn/)(XZ).The funders had no role in study design, data collection and analysis, decision to publish, or preparation of the manuscript.

## Conclusions

In European population older than 60 years, genetically predicted higher levels of resistin were associated with higher TB-BMD. A significant causality between resistin levels on BMD at different sites, fracture in certain parts of the body, and BMD in four different age groups between 0–60 years of age was not found in our study.

## 1. Introduction

Osteoporosis represents a chronic metabolic bone disease clinically characterized by a decrease in bone mineral density (BMD), manifested by low bone mass and degradation of the microstructure of bone tissue, resulting in bone fragility and an increased risk of fracture [1, 2]. Presently, the globally accepted diagnostic criterion for osteoporosis is predicated upon a BMD T-score equal to or less than -2.5, typically ascertained through dual-energy x-ray absorptiometry (DXA) [1]. Within this framework, a BMD T-score within the range of -1 to -2.5 signifies a state of low bone mass or reduced bone density [3]. The global prevalence of osteoporosis is about 18.3%, and its incidence is gradually increasing as the global population ages, constituting one of the biggest public health problems [4]. Osteoporotic fractures can contribute to a diminished quality of life and necessitate hospitalization, resulting in substantial economic losses [5]. There are more than 8.9 million osteoporotic fractures worldwide [6]. In the United States, approximately 1.5 million fractures are caused by osteoporosis each year, which will cost the U.S. healthcare system at least $5–10 billion annually, and by 2040, the associated costs will increase by 100–200% [7, 8].

The conventional wisdom is that obesity is a protective factor for BMD and fractures occurrence, but a growing body of evidence leads to the opposite conclusion [9]. Adipose tissue may be involved in the pathophysiologic process of osteoporosis by regulating bone metabolism [10]. It is not only an organ for storing energy, but also secretes a variety of biologically active molecules known as adipocytokines, such as adiponectin, leptin, and resistin [11]. The role of human resistin and its corresponding receptors remains poorly understood, and the operative mechanisms through which resistin exerts its effects in human physiology also incompletely elucidated [12, 13]. Differences in the origin, structure and function of resistin in humans and rodents increase the difficulty of studying human resistin in the population [14–16]. Currently, resistin has been described as a biomarker that may correlate with BMD and correlation studies between the two are gaining attention [17]. Several studies have reported that resistin levels have a potential influence on bone metabolism, but there are also findings that show no correlation between the two [18]. These results remain controversial and the causal relationship between resistin levels and BMD, fracture occurrence is unclear.

Mendelian randomization (MR) studies, as a research method within the realm of epidemiological research, offer a means to explore the causal linkage between resistin levels and BMD, as well as fracture susceptibility, facilitating causal inference with minimal avoidance of reverse causation and confounders (e.g., effects of lifestyle, dietary habits) [19]. It can be an alternative study to randomized controlled trials (RCTs) by examining the causal relationship between exposure (resistin levels) and outcome (BMD and fracture) based on published genome wide association studies (GWASs) data without the time, financial, human, and ethical constraints associated with RCTs [20].

## 2. Materials and methods

### 2.1 Overview of the study design

The two-sample MR study had to satisfy the following three assumptions: (1) Single nucleotide polymorphisms (SNPs) are strongly correlated with resistin levels; (2) the IVs do not interfere outcomes through other means than the resistin levels; (3) the instrumental variables (IVs) are not related to confounders [19]. All of our data came from the online database, the Integrative Epidemiology Unit (IEU) Open GWAS project (https://gwas.mrcieu.ac.uk/), thus no additional ethical approval was deemed necessary. The general design of the present study is shown in S1 Fig, and a flowchart of how the analysis will proceed step by step is shown in S2 Fig.

### 2.2 Instrumental variables acquisition

SNPs related to resistin levels were derived from a GWAS of 21,758 European populations, which is the most recent and largest sample size available [21]. We can also download from the online resources of the Systematic and Combined AnaLysis of Olink Proteins (SCALLOP) Consortium's (www.scallop-consortium.com). We set the *P*-value of each SNP from resistin levels to be less than $5 \times 10^{-8}$ to ensure its strong association with exposure and clumped the genetic variants within 10000kb at the threshold of linkage disequilibrium (LD) $r^2 < 0.001$. Ultimately, 13 SNPs relevant to resistin levels were screened as IVs.

### 2.3 BMD and fracture

In order to draw more comprehensive and reliable conclusions regarding the causative influence of resistin levels on BMD and fracture occurrence, we selected multiple sites for BMD [Total body bone mineral density (TB-BMD), Heel bone mineral density (HE-BMD), Ultradistal forearm bone mineral density (UF-BMD), Forearm bone mineral density (FA-BMD), Femoral neck bone mineral density (FN-BMD), Lumbar spine bone mineral density (LS-BMD)] and fractures (ankle fracture, arm fracture, leg fracture, spine fracture, wrist fracture). TB-BMD (n = 56,284) [22], FA-BMD (n = 8,143), FN-BMD (n = 32,735), and LS-BMD (n = 28,498) can also be found in the GEnetic Factors for Osteoporosis (GEFOS) Consortium website (http://www.gefos.org/) [23]. HE-BMD (n = 426,824) is simultaneously available for download on UK-Biobank ((http://www.nealelab.is/uk-biobank) [24]. Since the distal radius is a frequent site of fracture, we included UF-BMD (n = 21,907) as a separate site in the analysis [25]. Five sites of fracture were utilized to estimate the causal relationship between resistin levels and fracture [ankle fracture (n = 460,340), arm fracture (n = 460,340), leg fracture (n = 460,340), spine fracture (n = 460,340), wrist fracture (n = 460,340)]. GWAS summary statistics for fracture can also be retrieved from the Medical Research Council Integrative Epidemiology Unit (MRC-IEU) website (http://www.bristol.ac.uk/integrative-epidemiology/). To eliminate the impact of different ethnicities on the results, the GWAS for both BMD and fractures were all obtained from European populations. For an age-stratified analysis of resistin levels on TB-BMD, we chose a meta-analysis of GWAS in 5 age strata. Ages ranged from 0 to 60+, with each age group spanning 15 years [age 0–15 (n = 11,807), age 15–30 (n = 4,180), age 30–45 (n = 10,062), age 45–60 (n = 18,805), age > 60 (n = 22,504), and 86% of the participants were from the European population [22]. The characterization of the GWAS data information on exposures and outcomes is shown in S1 Table.

## 2.4 Statistical analysis

We first harmonized exposure (resistin levels) and outcome (BMDs and fractures). To ascertain that the contribution of a SNP to the exposure and the effect of the same SNP on the outcome corresponded to the same allele, we rectified ambiguous SNPs with incongruent alleles and palindromic SNPs with ambiguous strands, and those that could not be corrected would be excluded from the harmonization process [26]. Because not all exposed SNPs were available in the outcomes, and ambiguous SNPs with incongruent alleles that could not be corrected, as well as palindromic SNPs with ambiguous strands, were excluded from the analyses, the SNPs that are ultimately used in the MR analyses will not necessarily include all of the SNPs that were used as IVs.

To assess the strength of the IVs, we calculated the *F*-statistic $[F = R^2(N - K - 1/K(1 - R^2))]$ which is associated with the percentage of variability in the phenotype explained by the genetic variants ($R^2$) $[R^2 = 2 \times MAF \times (1 - MAF) \times \beta^2]$ [27]. $R^2$ is the exposed variance interpreted by the selection of IVs, and we get the vales of $R^2$ in the MR Steiger directionality test; N is the sample size; and K is the number of IVs. β is the coefficient per 1 allele and MAF (minimum allele frequency) is the minor allele frequency [28]. If the *F*-statistic is much larger than 10, then we consider it highly unlikely that the IVs are weakly biased [29]. Details of the exposed (clumped) SNPs are listed in the S2 Table. Thirteen SNPs were selected as IV with an *F*-statistic of 67.23 (S3 Table), indicating that the IV accurately predicted the outcome.

Since it is not possible that all genetic variation is a valid IV, three different approaches of MR [inverse variance weighted (IVW), MR-Egger, and weighted median] were applied to deal with variation heterogeneity and the pleiotropic effect. Random-effect IVW is generally considered as the major method in our MR analysis for testing the causality between exposure and the outcome because IVW can return the unbiased estimation of causal effects in the absence of pleiotropy [26]. Weighted median permits the usage of invalid IVs under the hypothesis that at least half of the IVs analysis are valid [30]. MR-Egger permits all genetic variants to have pleiotropic effect but it must have nothing to do with the correlation of variant-exposure [29, 31]. Although MR-Egger and weighted median using wider CI which leads to being less efficient, they could provide more steady estimates in a wider range of scenarios. They are therefore used to assess the robustness of the IVW results [31]. We draw scatter plots of the three methods in each MR to make the results more intuitive.

To assess the interference of potential confounders on causal inference of exposure and outcome, we performed multivariable MR (MVMR) considering traditional risk factors that may contribute to the outcome [32]. IV strength was similarly assessed by *F*-statistic and potential horizontal pleiotropy was detected by intercepts from MR-Egger regression. MVMR used two overlapping sets of SNPs to assess possible confounders and the effect of the exposure of interest on the same outcome. The aim was to assess whether the exposure of interest was a direct effect on outcome independent of other confounding factors that might have an effect on outcome [33]. We similarly set a significance threshold of $5 \times 10^{-8}$ for the two groups of GWAS included at a time, clumped at $r^2 = 0.01$ within 10,000 kb. Statistical analyses were performed in the R software (v 4.3.0) with "Two Sample MR" package (v 0.5.6).

## 2.5 Sensitivity analysis

We employed Cochran's Q test to examine whether heterogeneity was present in our study. MR pleiotropy residual sum and outlier (MR-PRESSO) test and MR-Egger regression were performed to detect the presence of horizontal pleiotropy. MR-PRESSO consists of a test for horizontal pleiotropy (MR-PRESSO global test), a test for correcting horizontal pleiotropy by

removing outliers (MR-PRESSO outlier test), and a test for correcting for significant differences in causal estimates before and after an outlier (MRPRESSO distortion test). If the MR-PRESSO global test results were significant, we corrected for horizontal pleiotropy by removing outliers and then ran MR-PRESSO again after the second MR analysis to judge whether horizontal pleiotropy had been eliminated. To assess whether the causal effect was driven by a particular SNP, we performed a "leave-one-out" analysis. In addition, to avoid the bias of reverse causation on the results, we evaluated the hypothesis of a potential causal association between resistin levels on BMD and resistin on fracture occurrence by the MR Steiger directionality test [34].

## 3. Results

### 3.1 Resistin levels and BMD

A total of 71 SNPs were used for MR analysis of resistin levels and BMD at different sites, of which 13 were TB-BMD, 13 were HE-BMD, 9 were UF-BMD, 12 were FA-BMD, 12 were FN-BMD, and 12 were LS-BMD. After we excluded outlier variants, we found that no significant causal relationship existed between resistin levels and BMDs at these sites. Then we performed MR-PRESSO outlier test again and no horizontal pleiotropy was found. The MR estimation of different methods for evaluating the causality of resistin levels on BMDs (after removing outliers) were displayed in Table 1. The scatter plots, leave-one-out analysis plots and funnel plots were presented in Fig 1, S3 and S6 Figs. The MR Steiger test results in TRUE, providing confidence in the direction of causal inference, but LS-BMD was found to be potentially pleiotropic in the MR-Egger intercept test (S4 Table). Heterogeneity was not revealed in this analysis and MR-PRESSO did not indicate the existence of horizontal pleiotropy (S4 Table).

### 3.2 Resistin levels and fracture

Fractures at different sites were used as outcome, with3 SNPs for ankle fracture, 3 SNPs for arm fracture, 3 SNPs for leg fracture, 3 SNPs for spine fracture, and 3 SNPs for wrist fracture. No causal relationship between resistin levels and fractures was obtained for IVW results in MR analysis of fractures at all sites (Table 2). Horizontal pleiotropy and heterogeneity were not detected (S4 Table). MR Steiger testing did not reveal the possibility of reverse causation (S4 Table). The scatter plots, leave-one-out analysis plots and funnel plots were shown in Fig 2, S4 and S7 Figs.

**Table 1. MR analysis between resistin levels and BMDs after removing all the outlier SNPs.**

| BMDs | nsnp | IVW | | Weighted median | | MR-Egger | |
|---|---|---|---|---|---|---|---|
| | | β(95%CI) | pval | β(95%CI) | pval | β(95%CI) | pval |
| Resistin levels and TB-BMD | 13 | 0.037(-0.008, 0.083) | 0.103 | 0.033(-0.026, 0.092) | 0.276 | 0.071(-0.014, 0.157) | 0.132 |
| Resistin levels and HE-BMD | 13 | 0.007(-0.006, 0.020) | 0.275 | 0.005(-0.011, 0.021) | 0.538 | 0.013(-0.009, 0.035) | 0.273 |
| Resistin levels and UF-BMD | 9 | -0.01(-0.120, 0.100) | 0.857 | -0.003(-0.154, 0.147) | 0.967 | -0.082(-0.593, 0.429) | 0.762 |
| Resistin levels and FA-BMD | 12 | -0.048(-0.156, 0.060) | 0.386 | -0.024(-0.173, -0.126) | 0.756 | 0.064(-0.128, 0.256) | 0.53 |
| Resistin levels and FN-BMD | 12 | 0.0087(-0.045, 0.062) | 0.760 | -0.005(-0.079, 0.070) | 0.905 | 0.091(-0.005, 0.188) | 0.094 |
| Resistin levels and LS-BMD | 12 | -0.053(-0.118, 0.012) | 0.111 | -0.003(-0.089, 0.083) | 0.946 | 0.064(-0.049, 0.177) | 0.292 |

**Abbreviation:** IVW, inverse variance weighted; TB-BMD, total body bone mineral density; HE-BMD, Heel bone mineral density; UF-BMD, Ultradistal forearm bone mineral density; FA-BMD, Forearm bone mineral density; FN-BMD, Femoral neck bone mineral density; LS-BMD, Lumbar spine bone mineral density; MR, mendelian randomization; SNP, single nucleotide polymorphism.

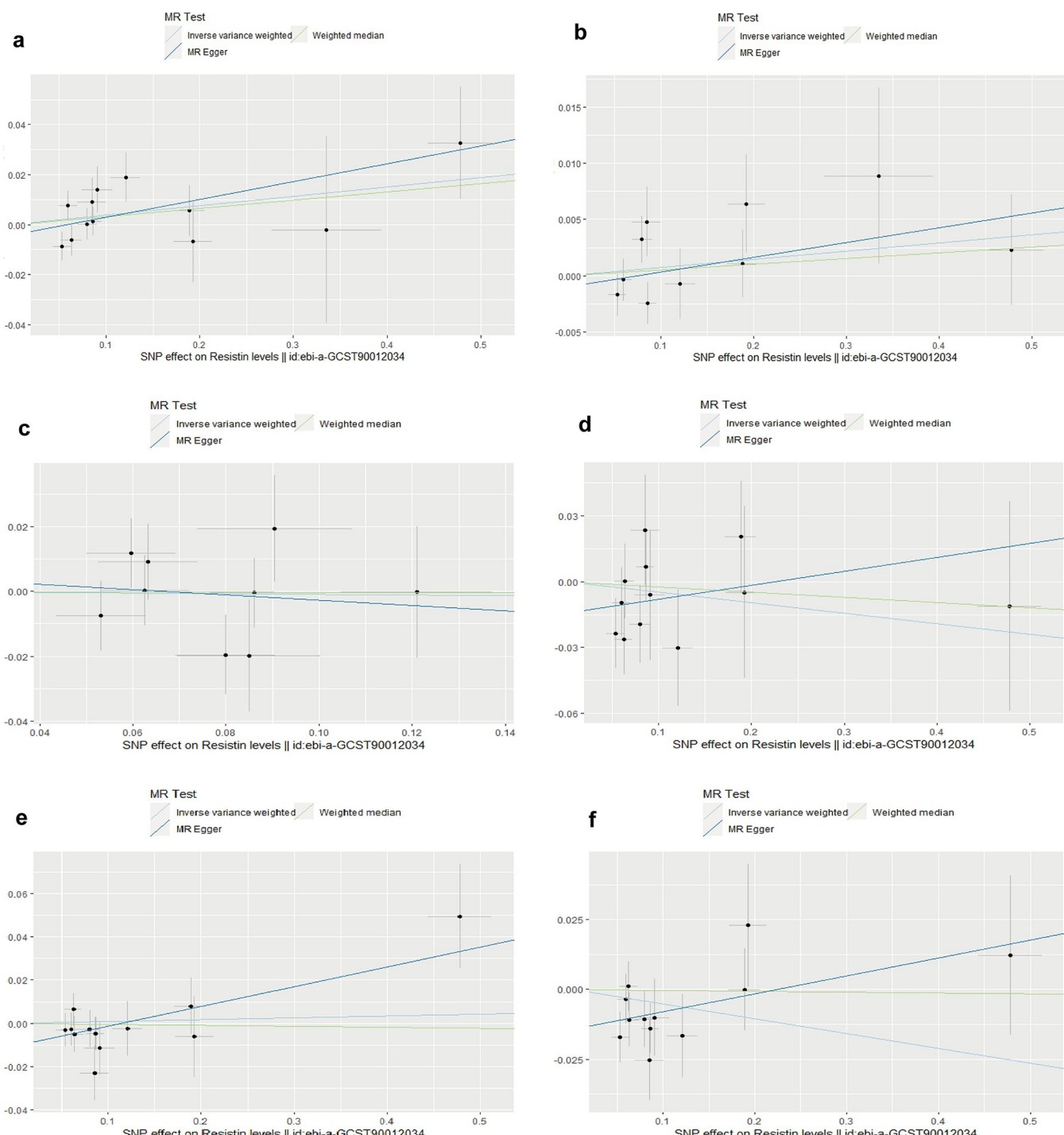

**Fig 1. Scatter plots for MR analyses of the causal effect of resistin levels on BMDs (after removing all the outlier SNPs).** (**a**) TB-BMD; (**b**) HE-BMD; (**c**) UF-BMD; (**d**) FA-DMD; (**e**) FN-BMD; (**f**) LS-BMD. MR, mendelian randomization; BMD, bone mineral density; SNP, single nucleotide polymorphisms; TB-BMD, total body bone mineral density; HE-BMD, Heel bone mineral density; UF-BMD, Ultradistal forearm bone mineral density; FA-BMD, Forearm bone mineral density; FN-BMD, Femoral neck bone mineral density; LS-BMD, Lumbar spine bone mineral density.

**Table 2. MR analysis between resistin levels and fracture at different sites.**

| Fracture | nsnp | IVW | | Weighted median | | MR-Egger | |
|---|---|---|---|---|---|---|---|
| | | β(95%CI) | pval | β(95%CI) | pval | β(95%CI) | pval |
| Resistin levels and Ankle fracture | 10 | -1.321E-03(-0.0038, 0.0011) | 0.287 | -1.573E-03(-0.0046, 0.0015) | 0.310 | 1.106E-03(-0.0057, 0.0015) | 0.757 |
| Resistin levels and Arm fracture | 10 | 4.060E-06(-0.0019, 0.0019) | 0.997 | 4.619E-04(-0.0020, 0.0029) | 0.709 | 1.175E-03(-0.0038, 0.0062) | 0.658 |
| Resistin levels and Leg fracture | 7 | -4.551E-04(-0.0023, 0.0014) | 0.636 | -1.705E-05(-0.0024, 0.0023) | 0.989 | -5.710E-03(-0.0158, 0.0044) | 0.321 |
| Resistin levels and Spine fracture | 5 | -5.059E-04(-0.0018, 0.0008) | 0.446 | -2.016E-04(-0.0018, 0.0014) | 0.803 | 9.415E-04(-0.0054, 0.0073) | 0.791 |
| Resistin levels and Wrist fracture | 11 | -8.609E-04(-0.0037, 0.0020) | 0.554 | -5.188E-04(-0.0039, 0.0029) | 0.766 | -3.759E-04(-0.0079, 0.0071) | 0.924 |

### 3.3 Resistin levels and TB-BMD at different groups between 0 and 60 years old

In order to know if there is a correlation between resistin levels and TB-BMD at different ages, we used TB-BMD at five ages [age 0–15 (13 SNPs), age 15–30 (11 SNPs), age 30–45 (13 SNPs), age 45–60 (13 SNPs), age > 60 (13 SNPs)]. In age-stratified analyses of resistin levels and TB-BMD, after removing all the outlier SNPs, we discovered that The IVW results show a positive causal relationship between resistin levels and TB-BMD above 60 years of age [β(95%CI): 0.093(0.021, 0.165), $P = 0.011$], and that the higher the level of resistin, the higher the trend of TB-BMD for those above 60 years of age. The results of Weighted median [β(95%CI): 0.111 (0.067, 0.213), $P = 0.035$] and MR-Egger [β(95%CI): 0.162(0.025, 0.2983), $P = 0.040$] further validate the results of IVW (Table 3 and Fig 3). Cochran's Q test, MR-Egger regression and MR-PRESSO did not suggest the presence of heterogeneity and horizontal pleiotropy (S4 Table). The result of the MR Steiger also supported our conclusion regarding the potential causal effect of resistin levels on TB-BMD in people over 60 years of age (S4 Table). Plots of the leave-one-out analysis demonstrated that there was no significant influential SNP driving the causal link and our conclusion was of stability. Nevertheless, after Bonferroni correction ($P < 0.05/5 = 0.01$), we consider it to be suggestive evidence of a potential causal relationship [35]. No results with causality were found in the MR analyses of other age groups (age 0–15, age15–30, age 30–45, age 45–60) as endpoints (Table 3 and Fig 3). The leave-one-out analysis plots and funnel plots were shown in S5 and S8 Figs.

### 3.4 Multivariable mendelian randomization analysis

To determine whether resistin levels are a direct causal factor for TB-BMD over 60 years of age, we performed a further MVMR. The effect of genetically predicted resistin levels on TB-BMD over 60 years of age persisted after accounting for factors such as physical activity, previous smoking, alcohol consumption, and Serum 25-Hydroxyvitamin D levels (Fig 4), with details of confounders shown in S5 Table. The different MR models were directionally consistent, the *F*-statistic did not detect the presence of weak instrumental bias, and the MR-Egger regression intercept did not detect potential horizontal pleiotropy.

## 4. Discussion

In this study, we evaluated whether there was a causal relationship between resistin levels and BMD, fracture occurrence using a two-sample MR. Our results show suggestive evidence of a potential positive causality between resistin levels and TB-BMD in people over 60 years of age [35]. Nevertheless, no causal relationship was found for resistin levels with bone mineral density and fractures in certain parts of the body in our present study. As no GWAS data have been published for fractures in different age groups, further MR studies are necessary to

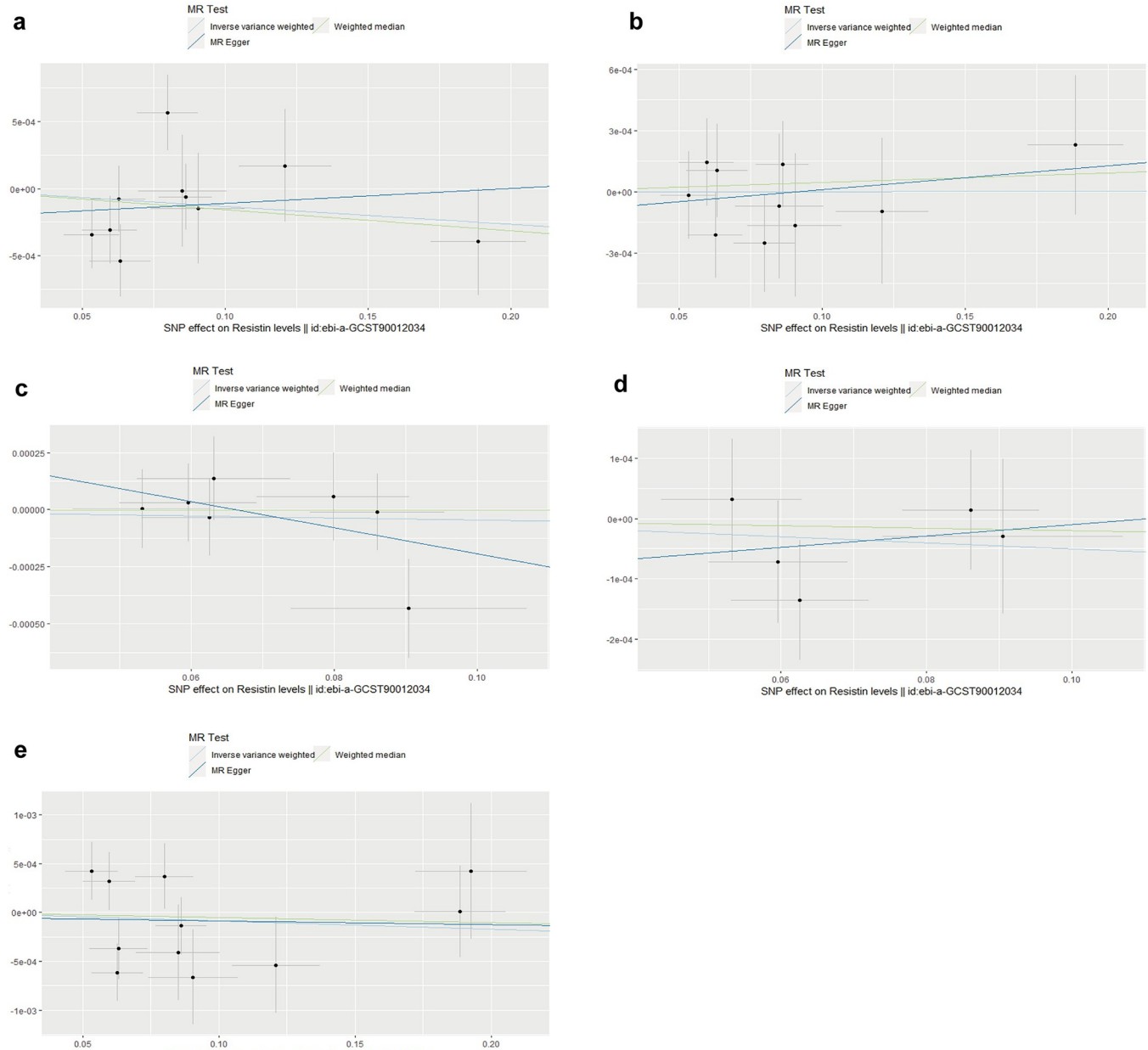

**Fig 2. Scatter plots for MR analyses of the causal effect of resistin levels on fracture. (a)** ankle fracture; **(b)** arm fracture; **(c)** leg fracture; **(d)** spine fracture; **(e)** wrist fracture.

explore the causal assessment of age stratification of fractures by resistin levels, and in particular the effect on fracture in European populations older than 60 years.

Resistin, initially identified within murine models, has been described as a cysteine-rich adipose tissue-specific secreted factor [36, 37], and was once named FIZZ3 because it had been found in inflammatory zone 3 of the lungs of asthmatic mice [38]. Resistin gene expression was found almost solely in white adipocytes and blood cells [39]. In the murine models, resistin is thought to impair glucose homeostasis, leading to glucose intolerance and insulin resistance [40, 41]. However, discernible disparities exist between human and rodent resistin with regard to genetic composition, protein structure, regulation of expression, protein expression

**Table 3. MR analysis between resistin levels and age stratified analysis of TB-BMD after removing outliers.**

| age stratified | nsnp | IVW | | Weighted median | | MR-Egger | |
|---|---|---|---|---|---|---|---|
| | | β(95%CI) | pval | β(95%CI) | pval | β(95%CI) | pval |
| Resistin levels and TB-BMD (age 0–15) | 13 | -0.023(-0.135, 0.089) | 0.689 | 0.007(-0.146, 0.161) | 0.923 | -0.018(-0.249, 0.213) | 0.880 |
| Resistin levels and TB-BMD (age 15–30) | 11 | 0.069(-0.134, 0.272) | 0.504 | 0.146(-0.116, 0.408) | 0.276 | 0.261(-0.292, 0.814) | 0.379 |
| Resistin levels and TB-BMD (age 30–45) | 13 | 0.062(-0.070, 0.193) | 0.357 | 0.067(-0.086, 0.220) | 0.390 | -0.010(-0.258, 0.237) | 0.936 |
| Resistin levels and TB-BMD (age 45–60) | 13 | 0.049(-0.042, 0.140) | 0.293 | 0.068(-0.040, 0.176) | 0.219 | 0.071(-0.107, 0.250) | 0.451 |
| Resistin levels and TB-BMD (age over 60) | 13 | 0.093(0.021, 0.165) | 0.011 | 0.111(0.008, 0.213) | 0.035 | 0.162(0.025, 0.298) | 0.040 |

sites, protein structure, and function [14]. Resistin in the human circulation is mainly derived from peripheral blood monocytes, macrophages and bone marrow cells [15]. Moreover, at the amino acid level, only 59% of the sequence of human resistin is identical to that of mouse resistin [16]. This may be one reason for the small number of studies and inconsistent conclusions regarding the effects of resistin on bone metabolism [13, 42].

The relationship between resistin and BMD in observational studies has been a topic of controversy. Mohiti-Ardekani et al. conducted an investigation of fasting serum adipokines and BMD in osteoporotic patients and non-osteoporotic controls and found that fasting plasma resistin levels in the osteoporotic group were remarkably related to femoral BMD, but not to LS-BMD ($P = 0.048$; $P = 0.56$) [18]. Nevertheless, in a cross-sectional study, resistin was described as a remarkable negatively relevant independent predictor of LS-BMD ($P < 0.001$) [43]. In addition, in a systematic review and meta-analysis of 59 cohort studies, there was no convincing data to favor a link for resistin and BMD [44]. The interesting thing is that Lee, in his latest meta-analysis, came up with different results, suggesting that the relevance of resistin with BMD was slightly stronger in the European male population than in other regions [17]. Whereas in postmenopausal women, the relevance of resistin with BMD was positive in European but not in Asians, suggesting that regional stratification is necessary [17]. Nevertheless, they included patients over 18 years of age and ignored subgroup analyses for age. A cross-sectional study by Bilha et al. showed that resistin was not linked to BMD, but due to the increased risk of comorbidities in people over 65 years of age, this population was not included in the study [45].

Resistin is thought to induce bone remodeling through the NF-κB pathway, an important pathway that plays a key role in osteoclastogenesis [46]. In a mechanistic study related to resistin and BMD, Thommesen et al. found that resistin could affect bone metabolism and remodeling by enhancing osteoblast differentiation and osteoclast recruitment, but did not elucidate the positive or negative effects of resistin on BMD [46]. The results of an *in vitro* study with comparatively similar results showed that recombinant resistin induced weak differentiation of preosteoblasts to osteoblasts while increasing osteoclast production [47]. In a recent study, Shang et al. demonstrated that resistin is highly expressed during osteogenic differentiation (OD) of bone marrow mesenchymal stem cells (BMSCs), and resistin can promote OD of BMSCs through activation of the PI3K/AKT/mTOR signaling pathway [48]. Furthermore, upregulation of resistin promotes OD by targeting transcription co-activators with PDZ-binding motifs (TAZ), and local injection of resistin dramatically promotes bone reparation and improves osteogenesis in a rat model of femoral condylar bone defects [48]. The results of this study elaborated on the positive effects of resistin on bone formation, but only reported the results of experiments in rats.

MR studies are less likely to be affected by confounding factors such as the environment and provide a more robust views of the causality linking risk factors and disease results than classical epidemiologic studies. Moreover, because an individual's genotype is determined at

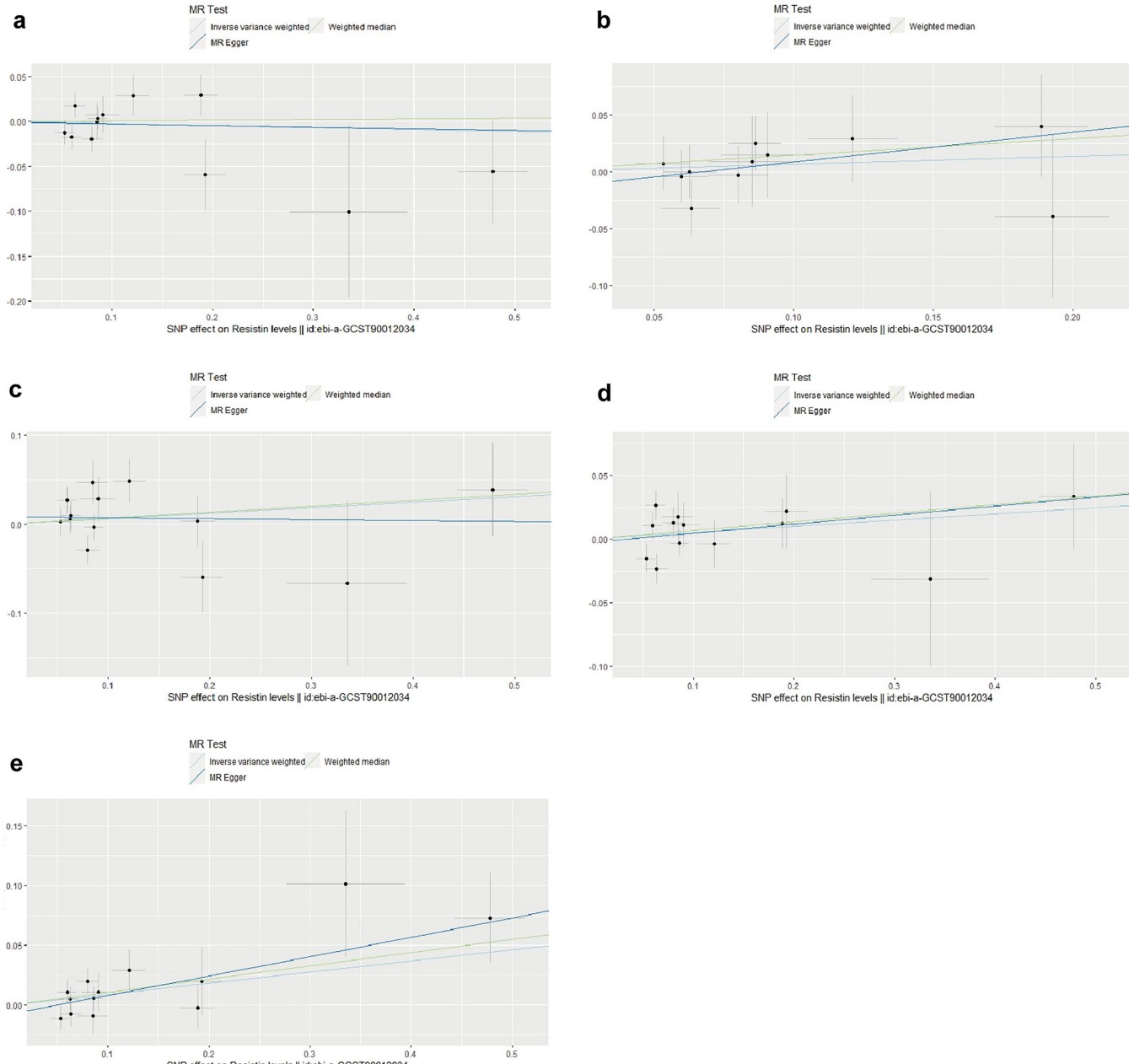

**Fig 3. Scatter plots for MR analyses of the causal effect of resistin levels on age stratified analysis of TB-BMD (after removing all the outlier SNPs). (a)** age 0–15; **(b)** age 15–30; **(c)** age 30–45; **(d)** age 45–60; **(e)** age > 60.

the formation of a fertilized egg and unable to be altered by following disease events, the order of causality is always from the genetic variant to the characterization of concern, thus erasing the possibility of inverse causation [49]. Although the methodology of MR studies is reliable, just as no research protocol is perfect, our study has several limitations. First, because the GWAS data only address indicators for individuals of European ancestry, our results are not entirely reflective of all ethnic populations. Consequently, caution should be exercised in applying our findings to ethnically diverse populations. Secondly, while the MR PRESSO global test in the analysis of resistin levels and LS-BMD did not detect significant horizontal

| Adjustment | No.of SNP | Method | β(95% CI) | β(95% CI) | P | Egger intercept P | F-statistic |
|---|---|---|---|---|---|---|---|
| physical activity | 17 | IVW | | 0.094 (0.021, 0.167) | 0.012 | 0.262 | 43.406 |
| | | Weighted median | | 0.087 (-0.035, 0.21) | 0.162 | | |
| | | MR Egger | | 0.155 (0.041, 0.27) | 0.008 | | |
| smoking: previous | 29 | IVW | | 0.076 (0.004, 0.148) | 0.038 | 0.074 | 29.006 |
| | | Weighted median | | 0.073 (-0.045, 0.191) | 0.227 | | |
| | | MR Egger | | 0.123 (0.035, 0.212) | 0.006 | | |
| alcohol consumption | 19 | IVW | | 0.105 (0.032, 0.177) | 0.005 | 0.178 | 45.399 |
| | | Weighted median | | 0.124 (0.004, 0.244) | 0.042 | | |
| | | MR Egger | | 0.169 (0.035, 0.302) | 0.013 | | |
| Serum 25−Hydroxyvitamin D levels | 13 | IVW | | 0.096 (0.024, 0.168) | 0.009 | 0.554 | 12.510 |
| | | Weighted median | | 0.104 (-0.022, 0.231) | 0.106 | | |
| | | MR Egger | | 0.092 (0.019, 0.165) | 0.014 | | |

-0.2    0    0.2    0.4

**Fig 4. Forest plot of MVMR considering common confounders of TB-BMD over 60 years of age.** Results of the causal assessment of resistin levels on TB-BMD over 60 years of age after adding one confounder at a time. MVMR, multivariable mendelian randomization; IVW, inverse variance weighted; CI, confidence interval.

pleiotropy (global *P* = 0.3675), and no significant outliers were detected either, but the MR-Egger regression detected the presence of suggestive horizontal pleiotropy (*P* = 0.035). Therefore, we believe that there is an effect of potential horizontal pleiotropy. But we did not screen for genetic variants associated with confounders through the PhenoScanner database as in previous studies [50]. Because it does not always distinguish the horizontal from the vertical pleiotropy, only the first one is biased against MR studies, and the precise biological role of many gene variants is not known [35, 51]. Thirdly, the SNPs of fracture used for analysis were only three per group. Therefore, GWAS with a larger range of resistin levels should be used in the updated MR analysis of resistin levels with fractures.

## 5. Conclusion

To sum up, Our MR study supports resistin as a suggestively protective factor in TB-BMD in European populations older than 60 years. This will likely facilitate the discovery of potential preventive and therapeutic strategies for osteoporosis in European populations over 60 years of age. However, MR studies with less biased and more accurate GWAS summary data are necessary to further confirm our results. Meanwhile, more studies need to be conducted to investigate the biological mechanisms involved. Given the limitations of this study, further MR studies are necessary to estimate the effect of resistin levels on lumbar spine bone density and fractures in different age groups.

## Supporting information

**S1 Fig. Overall design of the present study.**
(TIF)

**S2 Fig. Flow chart about how the analysis will proceed step by step.**
(TIF)

**S3 Fig. Plots of "leave-one-out" analyses for MR analyses of the causal effect of resistin levels on BMDs. (a)** TB-BMD; **(b)** HE-BMD; **(c)** UF-BMD; **(d)** FA-DMD; **(e)** FN-BMD; **(f)** LS-BMD.
(TIF)

**S4 Fig. Plots of "leave-one-out" analyses for MR analyses of the causal effect of resistin levels on fracture. (a)** ankle fracture; **(b)** arm fracture; **(c)** leg fracture; **(d)** spine fracture; **(e)** wrist fracture.
(TIF)

**S5 Fig. Plots of "leave-one-out" analyses for MR analyses of the causal effect of resistin levels on age stratified analysis of TB-BMD. (a)** age 0–15; **(b)** age 15–30; **(c)** age 30–45; **(d)** age 45–60; **(e)** age > 60.
(TIF)

**S6 Fig. Funnel plots for MR analyses of the causal effect of resistin levels on BMDs. (a)** TB-BMD; **(b)** HE-BMD; **(c)** UF-BMD; **(d)** FA-DMD; **(e)** FN-BMD; **(f)** LS-BMD.
(TIF)

**S7 Fig. Funnel plots for MR analyses of the causal effect of resistin levels on fracture. (a)** ankle fracture; **(b)** arm fracture; **(c)** leg fracture; **(d)** spine fracture; **(e)** wrist fracture.
(TIF)

**S8 Fig. Funnel plots for MR analyses of the causal effect of resistin levels on age stratified analysis of TB-BMD. (a)** age 0–15; **(b)** age 15–30; **(c)** age 30–45; **(d)** age 45–60; **(e)** age > 60.
(TIF)

**S1 Table. Characteristics of GWAS data of exposure and outcome.**
(DOCX)

**S2 Table. Detailed information of LD-independent SNPs (after clumping process) for exposure (resistin levels).**
(DOCX)

**S3 Table. *F*-statistics of instrumental variables (IVs).**
(DOCX)

**S4 Table. Sensitivity analysis outcome of all analyses (after removing outliers).**
(DOCX)

**S5 Table. Characterization of GWAS statistics for confounding factors.**
(DOCX)

## Author Contributions

**Conceptualization:** Xian Zhang.

**Data curation:** Taichuan Xu.

**Formal analysis:** Taichuan Xu, Xian Zhang.

**Funding acquisition:** Xian Zhang.

**Investigation:** Chao Li, Xian Zhang.

**Methodology:** Taichuan Xu, Chao Li.

**Project administration:** Xian Zhang.

**Resources:** Xian Zhang.

**Software:** Taichuan Xu.

**Supervision:** Xian Zhang.

**Validation:** Taichuan Xu, Yitao Liao.

**Visualization:** Yenan Xu, Zhihong Fan.

**Writing – original draft:** Taichuan Xu, Chao Li.

**Writing – review & editing:** Taichuan Xu, Yitao Liao.

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
