## [Decision Letter · Decision Letter 0]

20 Mar 2024

PONE-D-23-37938Is there a causal relationship between resistin levels and bone mineral density, fracture occurrence? A mendelian randomization studyPLOS ONE

Dear Dr. Zhang,

Thank you for submitting your manuscript to PLOS ONE. After careful consideration, we feel that it has merit but does not fully meet PLOS ONE’s publication criteria as it currently stands. Therefore, we invite you to submit a revised version of the manuscript that addresses the points raised during the review process.Please ensure that your decision is justified on PLOS ONE’s publication criteria and not, for example, on novelty or perceived impact.

We look forward to receiving your revised manuscript.

Kind regards,

Mukhtiar Baig, Ph.D.

Academic Editor

PLOS ONE

Journal Requirements:

Reviewers' comments:

Reviewer's Responses to Questions

**Comments to the Author**

1. Is the manuscript technically sound, and do the data support the conclusions?

Reviewer #1: Yes

2. Has the statistical analysis been performed appropriately and rigorously? 

Reviewer #1: Yes

3. Have the authors made all data underlying the findings in their manuscript fully available?

Reviewer #1: No

4. Is the manuscript presented in an intelligible fashion and written in standard English?

Reviewer #1: Yes

5. Review Comments to the Author

Reviewer #1: Xu’et al reported the potential causal effect of the resistin levels on bone mineral density over the age of 60. This article has some clinical value, but there are some limitations:

1. Why didn’t the author analyze the resistin levels' causal effect on fracture risks based on different age groups? For example, authors should analyze the causal associations of the resistin levels with fracture risks in different age groups (divided by the age of 60), because the higher resistin levels were significantly correlated with higher TB-BMD in the people over the age of 60.

2. The authors should evaluate the reverse causal associations between the exposure and the outcome such as the steiger test.

3. The authors didn’t evaluate the causal effects of exposures on potential confounding factors, which will affect the reliability of the results. Authors should screen some common confounding factors as outcomes and perform further Mendelian randomization analysis to identify the causal effects of exposures on these confounding factors.

4. Authors could list the detailed basic information of exposures and outcomes in a table, which will help make the article more concise and clearer.

6. PLOS authors have the option to publish the peer review history of their article (what does this mean?). If published, this will include your full peer review and any attached files.

Reviewer #1: No

---

## [Author Response · Author response to Decision Letter 0]

2 Apr 2024

Dear Editors and Reviewers, 

We would like to express our gratitude to the editors and reviewers for dedicating their time and effort to reviewing our paper entitled “Is there a causal relationship between resistin levels and bone mineral density, fracture occurrence? A mendelian randomization study” (Manuscript ID: PONE-D-23-37938). The insightful comments have been invaluable in improving the quality of our study. We have carefully revised the manuscript in accordance with the reviewers' suggestions and have taken care to minimize any grammatical errors. For editorial purposes, we have provided a marked-up copy of the manuscript called "Revised Manuscript with Track Changes" and an unmarked revised manuscript called "Manuscript".

We are deeply grateful for the editors' and reviewers' consideration and hope that this revised manuscript meets the requirements for publication in PLOS ONE. 

Responds to the reviewer’s comments:

COMMENT #1. Why didn’t the author analyze the resistin levels' causal effect on fracture risks based on different age groups? For example, authors should analyze the causal associations of the resistin levels with fracture risks in different age groups (divided by the age of 60), because the higher resistin levels were significantly correlated with higher TB-BMD in the people over the age of 60.

Response: We would like to thank the reviewer for this important comment. We fully agree that it is necessary to assess the causal effect of resistin levels on fractures in different age groups.

We did our best to retrieve data pertaining to fractures across various age groups from several prominent GWAS databases, such as the IEU Open GWAS project, GWAS Catalog, and FinnGen. Unfortunately, we found no relevant data published. In order to mitigate the risk of overlooking GWAS data that may have been published but not yet updated in the databases, we also searched MEDLINE for relevant GWAS for fractures in different age groups, but remained unimpressed. 

Therefore, we have added the relevant explanation in the first paragraph of the discussion (Lines 285-288, unmarked version), as well as a reference to it in the section on the limitations of the manuscript discussion (Lines 365-367, unmarked version). 

The revised sentences in the manuscript were as follows:

As no GWAS data have been published for fractures in different age groups, further MR studies are necessary to explore the causal assessment of age stratification of fractures by resistin levels, and in particular the effect on fracture in European populations older than 60 years.

Given the limitations of this study, further MR studies are necessary to estimate the effect of resistin levels on lumbar spine bone density and fractures in different age groups.

COMMENT #2. The authors should evaluate the reverse causal associations between the exposure and the outcome such as the steiger test.

Response: We would like to thank the reviewer for this important comment. We have added MR Steiger as a sensitivity analysis and the results are presented in S4 Table. The results of MR Steiger did not find evidence of reverse causation. We have added relevant descriptions of the use of the MR Steiger to the methods section of the abstract (Lines 33-34, unmarked version) and in the sensitivity analysis section (Lines 191-194, unmarked version) of the manuscript, as well as in the results section (Lines 206-208, unmarked version; Lines 223-224, unmarked version; Lines 251-253, unmarked version).

The revised sentences in the manuscript were as follows:

Simultaneously, heterogeneity and horizontal pleiotropy were also assessed and the direction of potential causality was detected by MR Steiger.

In addition, to avoid the bias of reverse causation on the results, we evaluated the hypothesis of a potential causal association between resistin levels on BMD and resistin on fracture occurrence by the MR Steiger directionality test[34].

References:

34. Hemani G, Tilling K, Davey Smith G. Orienting the causal relationship between imprecisely measured traits using GWAS summary data. PLoS Genet. 2017;13: e1007081. doi:10.1371/journal.pgen.1007081

The MR Steiger test results in TRUE, providing confidence in the direction of causal inference, but LS-BMD was found to be potentially pleiotropic in the MR-Egger intercept test (S4 Table).

MR Steiger testing did not reveal the possibility of reverse causation (S4 Table).

The result of the MR Steiger also supports our conclusion regarding the potential causal effect of resistin levels on TB-BMD in people over 60 years of age (S4 Table).

A table of the supporting information that was revised in the manuscript was as follows:

S4 Table. Sensitivity analysis outcome of all analyses (after removing outliers).(visible in file labeled "Response to Reviewers")

COMMENT #3. The authors didn’t evaluate the causal effects of exposures on potential confounding factors, which will affect the reliability of the results. Authors should screen some common confounding factors as outcomes and perform further Mendelian randomization analysis to identify the causal effects of exposures on these confounding factors.

Response: We would like to thank the reviewer for this important comment. To exclude the influence of confounding factors on the reliability of the results, we added a multivariable mendelian randomization analysis to assess whether the protective effect of resistin on total body BMD in European populations over the age of 60 was independent of confounding factors. 

We chose physical activity, previous smoking, alcohol consumption, and Serum 25-Hydroxyvitamin D levels as confounders. The results of the multivariable mendelian randomization analysis showed that the effect of resistin levels on total body BMD over the age of 60 years persisted after these confounding factors were excluded. 

We have added some description of multivariable Mendelian randomization to the manuscript. First, the use of multivariable Mendelian randomization is mentioned in the methods section of the abstract (Lines 34-36, unmarked version). The methodology of the multivariable mendelian randomization analysis is described in 2.4 Statistical analysis (Lines 169-178, unmarked version) and the results are presented in 3.4 multivariable mendelian randomization analysis (Lines 265-273, unmarked version). The forest plot for the multivariable mendelian randomization analysis is shown in Fig 4, and the GWAS data characterization for confounding factors is shown in S5 Table.

The revised sentences in the manuscript were as follows:

Multivariable MR (MVMR) analysis was used to identify whether confounding factors affected the reliability of the results.

To assess the interference of potential confounders on causal inference of exposure and outcome, we performed multivariable MR (MVMR) considering traditional risk factors that may contribute to the outcome[32]. IV strength was similarly assessed by F-statistic and potential horizontal pleiotropy was detected by intercepts from MR-Egger regression. MVMR used two overlapping sets of SNPs to assess possible confounders and the effect of the exposure of interest on the same outcome. The aim was to assess whether the exposure of interest was a direct effect on outcome independent of other confounding factors that might have an effect on outcome[33]. We similarly set a significance threshold of 5×10-8 for the two groups of GWAS included at a time, clumped at r2 = 0.01 within 10,000 kb.

References:

32. Sanderson E. Multivariable Mendelian Randomization and Mediation. Cold Spring Harb Perspect Med. 2021;11: a038984. doi:10.1101/cshperspect.a038984

33. Sanderson E, Davey Smith G, Windmeijer F, Bowden J. An examination of multivariable Mendelian randomization in the single-sample and two-sample summary data settings. Int J Epidemiol. 2019;48: 713–727. doi:10.1093/ije/dyy262

3.4 Multivariable Mendelian randomization analysis

To determine whether resistin levels are a direct causal factor for TB-BMD over 60 years of age, we performed a further MVMR. The effect of genetically predicted resistin levels on TB-BMD over 60 years of age persisted after accounting for factors such as physical activity, previous smoking, alcohol consumption, and Serum 25-Hydroxyvitamin D levels (Fig 4), with details of confounders shown in S5 Table. The different MR models were directionally consistent, the F-statistic did not detect the presence of weak instrumental bias, and the MR-Egger regression intercept did not detect potential horizontal pleiotropy.

The forest plot results of the added multivariable mendelian randomization analysis were as follows:

 (visible in file labeled "Response to Reviewers")

Fig 4. Forest plot of MVMR considering common confounders of TB-BMD over 60 years of age. Results of the causal assessment of resistin levels on TB-BMD over 60 years of age after adding one confounder at a time. MVMR, multivariable mendelian randomization; IVW, inverse variance weighted; CI, confidence interval.

A table characterizing the GWAS data for confounders was shown below:

S5 Table. Characterization of GWAS statistics for confounding factors. (visible in file labeled "Response to Reviewers")

COMMENT #4. Authors could list the detailed basic information of exposures and outcomes in a table, which will help make the article more concise and clearer.

Response: We would like to thank the reviewer for this valuable comment. The first version of the manuscript contained a table of basic information about exposures and outcome in S1 Table. We have adjusted the descriptions and table names in the text for clarity.

The revised table name was as follows:

S1 Table. Characteristics of GWAS data of exposure and outcome.

In addition, the manuscript has been revised by a native speaker. 

We re-formatted the manuscript with reference to PLOS ONE's style requirements. Moreover, the authors have revised the manuscript for parts that were considered illogical or poorly expressed, as can be seen in the manuscript titled “Revised Manuscript with Track Changes”.

We tried our best to improve the manuscript and made some changes in the manuscript. We appreciate for Editors/Reviewers’ warm work earnestly, and hope that the correction will meet with approval. Once again, thank you very much for your comments and suggestions.

Looking forward to hearing from you soon.

Thank you and best regards.

Sincerely,

Xian Zhang

Email: wxzy008@njucm.edu.cn

Tel: +8613616173567

Fax: +860510-82703775

Address: Wuxi Affiliated Hospital of Nanjing University of Chinese Medicine, No. 8, Zhongnan West Road, Lihu Street, Binhu District, Wuxi, Jiangsu, China.

---

## [Decision Letter · Decision Letter 1]

27 May 2024

Is there a causal relationship between resistin levels and bone mineral density, fracture occurrence? A mendelian randomization study

PONE-D-23-37938R1

Dear Dr. Zhang,

We’re pleased to inform you that your manuscript has been judged scientifically suitable for publication and will be formally accepted for publication once it meets all outstanding technical requirements.

Kind regards,

Mukhtiar Baig, Ph.D.

Academic Editor

PLOS ONE

Reviewer's Responses to Questions

**Comments to the Author**

1. If the authors have adequately addressed your comments raised in a previous round of review and you feel that this manuscript is now acceptable for publication, you may indicate that here to bypass the “Comments to the Author” section, enter your conflict of interest statement in the “Confidential to Editor” section, and submit your "Accept" recommendation.

Reviewer #1: (No Response)

2. Is the manuscript technically sound, and do the data support the conclusions?

Reviewer #1: (No Response)

3. Has the statistical analysis been performed appropriately and rigorously? 

Reviewer #1: (No Response)

4. Have the authors made all data underlying the findings in their manuscript fully available?

Reviewer #1: (No Response)

5. Is the manuscript presented in an intelligible fashion and written in standard English?

Reviewer #1: (No Response)

6. Review Comments to the Author

Reviewer #1: (No Response)

7. PLOS authors have the option to publish the peer review history of their article (what does this mean?). If published, this will include your full peer review and any attached files.

Reviewer #1: No

---

## [Editor Report · Acceptance letter]

15 Jul 2024

PONE-D-23-37938R1 

PLOS ONE

Dear Dr. Zhang, 

I'm pleased to inform you that your manuscript has been deemed suitable for publication in PLOS ONE. Congratulations! Your manuscript is now being handed over to our production team.

Kind regards, 

on behalf of

Professor Mukhtiar Baig 

Academic Editor

PLOS ONE